# A Typical Distributed Generation Scenario Reduction Method Based on an Improved Clustering Algorithm

**Sitong Lv [1], Jianguo Li [1,*], Yongxin Guo [2] and Zhong Shi [1]** 

[1]   College of Electrical Engineering, Shanghai Dianji University, Shanghai 20136, China;
      sitonglv@163.com (S.L.); shizhongsd@126.com (Z.S.)
[2]   Training Center of Jilin Province, State grid Corporation of China, Jilin 130062, China; dufufu34@163.com
[*]   Correspondence: lijiang@sdju.edu.cn; Tel.: +86-199-2126-6168

**Abstract:** In recent years, distributed generation (DG) technology has developed rapidly. Renewable energy, represented by wind energy and solar energy, has been widely studied and utilized. In order to give full play to the advantages of distributed generation and to meet the challenges of DG access to the power grid, the multi-scenario analysis method commonly used in DG optimal allocation method is studied in this paper. In order to solve the problems that may arise from using large-scale scenes in the planning of DG considering uncertainties by using multi-scene analysis method, the cluster analysis method suitable for large-scale scene reduction in scene reduction method is introduced firstly, and then an improved clustering algorithm is proposed. The validity of the scene reduction method is tested, and the feasibility of the reduction method is verified. Finally, the method mentioned in this paper is compared with other commonly used methods through IEEE-33 node system.

**Keywords:** distributed generation; multi-scene analysis; scene reduction; improved clustering algorithm

## 1. Introduction

With the rapid development of the global economy in the 21st century, the demand for energy in various countries is also increasing. The excessive use of traditional energy sources like petroleum and coal has caused serious environmental pollution. Under such circumstances, it is necessary to improve the development and utilization of existing non-renewable energy resources, develop and utilize new environmentally friendly energy resources, and provide necessary supplementary power and innovations to the existing energy system. Therefore, distributed generation (DG) has received extensive attention and support [1–3]. It is of great significance to vigorously develop distributed energy and give full play to the role of DG in the power grid [4–8].

At this stage, a lot of research work [9–11] has been done on the output of distributed generation considering uncertainties, including DG location and capacity, demand side response, network reconfiguration, power and voltage quality, etc. A control algorithm based on improved amplitude adaptive notch filter (AANF) is proposed for generation management of different energy sources in autonomous micro-grid. The main objective of the proposed modified AANF (MAANF)-based control algorithm is to control the flow of active and reactive powers among the different energy sources and the load, along with the regulation of the point of common coupling (PCC) voltages and mitigation of the source current harmonics [9]. The voltage-controlled oscillator (VCO) less phase-locked loop (PLL)-based control of voltage source converter was presented to improve power quality. Using the proposed control, the reactive power compensation, harmonics reduction and load balancing are carried out in the system [10]. A new attempt of utilizing the sunflower optimization (SFO) algorithm in solving the problem of optimal power flow (OPF) in the field of power systems was introduced in order

to optimize the generating units' fuel cost under the system constraints. The SFO algorithm is used to minimize the fitness function and yields the best solutions of the problem [11]. A decision-making algorithm that has been developed for the optimum size and placement of DG units in distribution networks. The algorithm that is very flexible to changes and modifications can define the optimal location for a DG unit (of any type) and can estimate the optimum DG size to be installed, based on the improvement of voltage profiles and the reduction of the network's total real and reactive power losses [12]. An economic study is carried out to analyze the economic feasibility for the integration of flywheel energy storage systems (FESS) with a wind power plant. It was concluded that the installation of the FESS is only feasible with the government subsidy in renewable energy projects, if considering that installation costs would not be reduced more than 10% of the estimated value [13]. The impact of three different types of distributed generation (diesel generator, wind turbine and photovoltaic (PV)) on distribution networks' voltage profile and power losses is studied. The obtained results show that different types of DG influence differently the distribution network and that their precise location and size are vital in reducing power losses and improving the voltage stability [14].

Wind power output and photovoltaic output have seasonal and diurnal periodicity [15,16]. Based on the daily and annual characteristics of wind and solar energy resources, a large number of scenarios need to be calculated and analyzed in order to comprehensively evaluate the feasibility and rationality of the planning and operation scheme. In power systems with large-scale distributed energy, if we can extract representative typical scenarios from a large number of historical resources data of distributed energy, then we can use typical scenario sets to reflect the changing characteristics of distributed energy in the cycle. This method is of great significance to the evaluation of distributed energy acceptance capacity, power planning, energy storage planning, operation planning and scheduling.

The existing scenario analysis methods can be divided into three categories: (1) scenario subtraction for day-ahead scheduling, which generates a large number of prediction scenarios based on scenario generation technology, and then reduces the large-scale prediction scenarios by scenario subtraction technology, and merges a large number of scenarios into a few typical scenarios [17]; (2) mid-and long-term power planning for the purpose. In the typical Japanese method, the method often chooses a day or a specific day which is close to the average value of the cycle as a typical day based on user experience and the research purposes [18]. (3) Time series simulation method for medium and long term power planning [19]. This method can characterize the time series variation characteristics of load/wind power output in the cycle, and provide realistic simulation scenarios and rich results for the whole network optimization of unit start-up and shutdown. Although time series simulation method has engineering application value, it generally needs to simulate the whole year time series, which has the problem of low computational efficiency.

Scenario analysis has been widely used in the optimal configuration of DG at this stage. In the process of scenario analysis, all historical data are used for multi-scenario analysis. Although the result of analysis is comprehensive, it will lead to a sharp increase in computational complexity and difficulty in solving problems. Therefore, it is necessary to reduce large-scale scenarios. In this paper, a scene reduction method based on improved clustering algorithm is proposed to reduce large-scale scenes in order to take into account both the efficiency and accuracy of computation and to form a scene reduction set that can reflect the data characteristics of the original scene set.

In view of the above analysis, this paper mainly studies the following work aspects:

(1)　The uncertainties of wind power and photovoltaic distributed energy are modeled. The characteristics of annual scenarios, seasonal scenarios, continuous multi-day scenarios and typical day scenarios are analyzed. The uncertainties and time series characteristics of the two types of distributed energy are analyzed.

(2)　In order to fully reflect the timing characteristics of two types of intermittent DGs and to avoid the difficulties caused by large-scale data in multi-scenario analysis. In this paper, a typical scenario set generation method based on improved clustering algorithm is proposed to reduce the wind

power and photovoltaic data in the computing cycle and form a typical scenario set that can reflect the historical data characteristics in the computing cycle.

(3) The proposed scenario reduction method based on improved clustering algorithm is validated by using the wind power output scenarios obtained from uncertain modeling.

## 2. Characteristics of Typical Intermittent Distributed Generation

### 2.1. Uncertainty Model of Wind Power Generation and Photovoltaic Power Generation

Wind energy resources are among the most abundant and mature intermittent distributed power sources. Wind power is affected by many factors, which can be roughly divided into atmospheric characteristics, terrain characteristics, wind power, behavior index, other indexes and geographical conditions. Wind power has more influence and is greatly influenced by the change of natural environment, which leads to the obvious uncertainty of wind power output.

In this paper, Weibull distribution with two parameters, which has the best application effect in engineering practice, is adopted. Its probability density function [20,21] is:

$$f(v) = \frac{k}{c}\left(\frac{v}{c}\right)^{k-1} exp\left[-\left(\frac{v}{c}\right)^{k}\right] \tag{1}$$

where $k$ is the shape parameters, $c$ is the scale parameters, and $v$ is the wind speed. The wind speed data used in this paper are from China National Meteorological Data Center.

Scale parameter $c$ and shape parameter $k$ can be determined by Equation (2):

$$k = \left(\frac{\sigma_w}{E_w}\right)^{-1.086}, \ c = \frac{\overline{v}}{\Gamma(1 + k^{-1})} \tag{2}$$

where $\sigma_w$ is the variance of $v$; $\overline{v}$ is the average value of $v$; $E_w$ is the generation capacity; $\Gamma$ is the gamma function. The turbine used here is a W2000-116-90 unit produced by the Shanghai Electric Group (Shanghai, China).

When the cut-in wind speed is reached, the turbine starts to produce its power. As the wind speed increases, the turbine output will also increase. When the wind speed is too high, in order to protect the turbine, the turbine equipment will be automatically removed. Therefore, the output power of the turbine can generally be expressed by a piecewise function, as shown in Equation (3):

$$P_{WTG(v)} = \begin{cases} 0, & 0 \leq v \leq v_{ci} \\ \frac{P_r(v - v_{ci})}{v_r - v_{ci}}, & v_{ci} < v \leq v_{cr} \\ P_r, & v_{cr} < v \leq v_{co} \\ 0, & v > v_{co} \end{cases} \tag{3}$$

where $P_{WTG(v)}$ is the turbine output, $v_{ci}$ is the cut-in wind speed, $v_{cr}$ is the rated wind speed, $v_{co}$ is the cut-out wind speed, and $P_r$ is the rated active power. The cut-in wind speed is 3 m/s, the cut-out wind speed is 25 m/s and the rated wind speed is 6.7 m/s.

Solar energy is the most abundant of all renewable energy sources. Photovoltaic power generation has remarkable flexibility, and its installation is simple and flexible. It is an important form and component of distributed power generation. At the same time, with the continuous development of photovoltaic power generation and the increase of investment, the cost of photovoltaic power generation has decreased significantly in recent years. The continuous increase of grid-connected photovoltaic power generation has also brought many impacts on the current grid scheduling and control, and this impact will continue to increase with the increase of grid-connected photovoltaic power generation. Similar to wind power, photovoltaic power generation has obvious randomness and uncertainty.

In this paper, Beta distribution in probability model is used to describe the uncertainty of illumination intensity. Its probability density function [22,23] is shown in Equation (4):

$$f(S) = \frac{\Gamma(\alpha+\beta)}{\Gamma(\alpha)\Gamma(\beta)}\left(\frac{S}{S_{max}}\right)^{\alpha-1}(1 - \frac{S}{S_{max}})^{\beta-1} \tag{4}$$

where $S$ is the illumination intensity; $S_{max}$ is the maximum illumination intensity; $\alpha$ and $\beta$ is the two parameters corresponding to Beta distribution. The illumination intensity data in this paper are from China National Meteorological Data Center.

$\alpha$ and $\beta$ can be calculated by the expected $\mu$ and variance $\sigma^2$ of illumination intensity over a certain period of time, as shown in Equations (5) and (6):

$$\alpha = \mu[\frac{\mu(1-\mu)}{\sigma^2} - 1] \tag{5}$$

$$\beta = (1-\mu)[\frac{\mu(1-\mu)}{\sigma^2} - 1] \tag{6}$$

The output power of photovoltaic power generation equipment will gradually increase with the increase of illumination intensity until it reaches the rated power. The relationship between output power and illumination intensity can be expressed by Equation (7):

$$P_{PVG} = \begin{cases} P_{PVG,r}S/S_r, & S \le S_r \\ P_{PVG,r}, & S > S_r \end{cases} \tag{7}$$

*2.2. Characteristic Analysis of Typical Intermittent Distributed Generation Scene*

Wind power output is affected by wind speed uncertainties, while photovoltaic output is mainly affected by illumination intensity uncertainties. Therefore, both wind power output and photovoltaic output have obvious volatility and randomness. This section will analyze the scene characteristics of two kinds of DGs and study their inherent characteristics.

2.2.1. Characteristic Analysis of Wind power Output Scene

According to the uncertain modeling results of wind speed, the corresponding wind power output curve can be obtained. Figure 1 is the annual wind power output variation curve of a certain area. In order to observe its variation more intuitively, the fitting curve is made. Figure 2 is the average wind power output variation curve in different seasons, Figure 3 is the continuous multi-day wind power output variation curve drawn randomly, and Figure 4 is the typical wind power output curve.

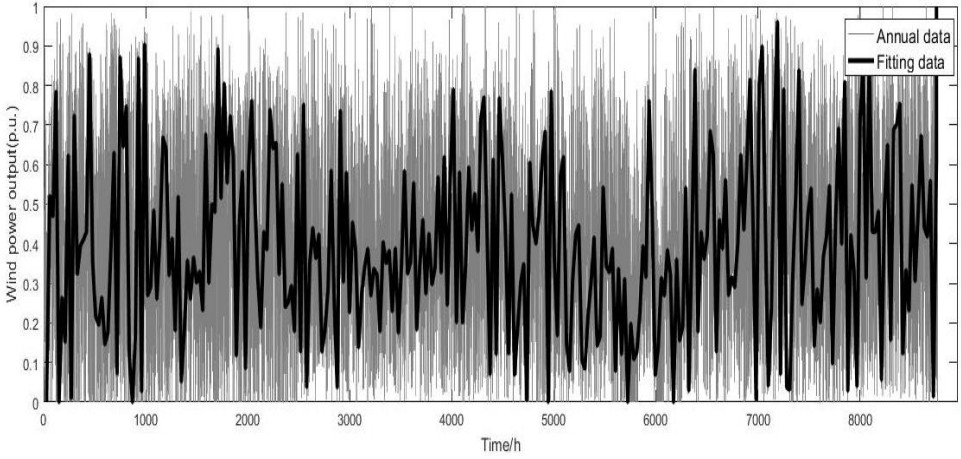

**Figure 1.** Annual wind power output curve.

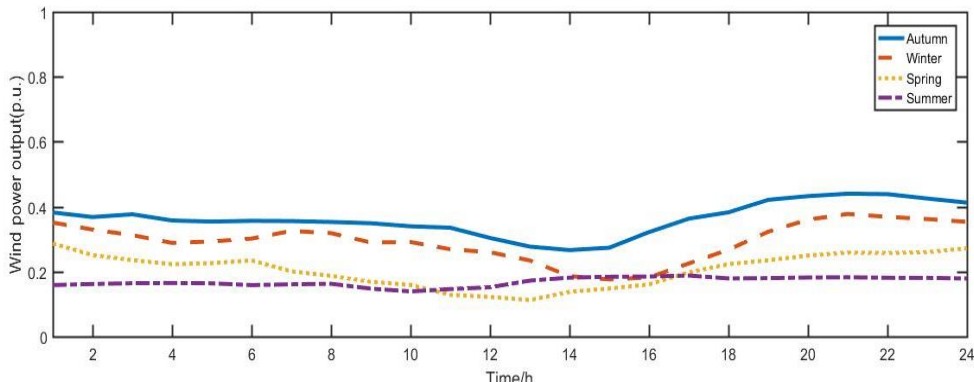

**Figure 2.** Wind power output curve in different seasons.

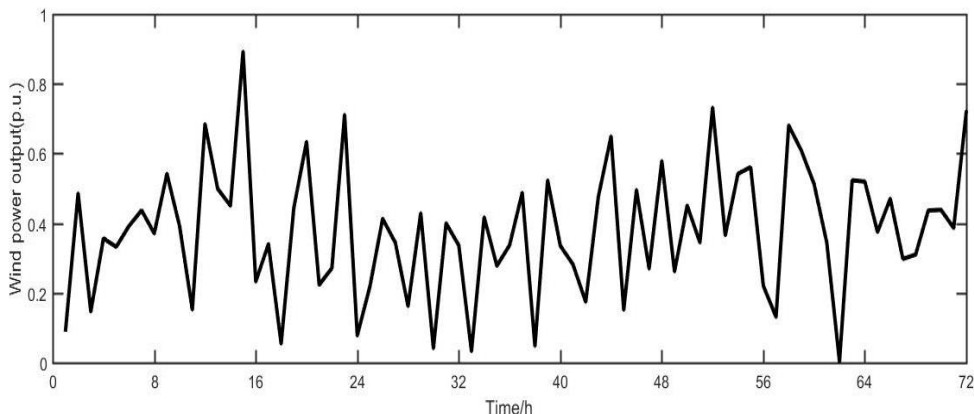

**Figure 3.** Continuous multi-day wind power output variation curve.

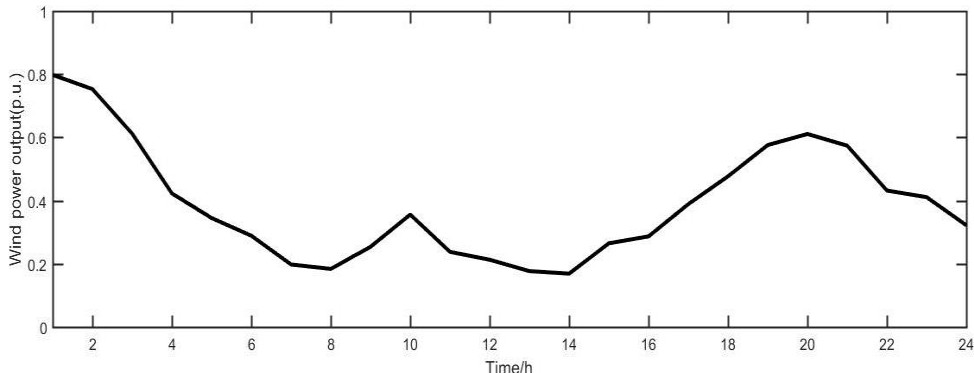

**Figure 4.** Typical wind power output curve.

The characteristics of the above wind power output curves are analyzed and summarized as follows:

(1) Random. As shown in Figure 1, the fitting curve of the annual wind power output curve can be clearly observed. For the hourly statistics of wind power output, the output at each moment shows obvious uncertainty.

(2) Intermittence. The wind speed has obvious intermittence, and is affected by the cutting-in speed and cutting-out speed of wind turbines, so there are some points in the curve where the output of wind turbines is zero. The point where the output of these turbines is zero may be due either to the failure to reach the cut-in wind speed or to the fact that the turbine has been removed because the cut-out wind speed has been reached, thus the output of wind power is not continuous.

(3) Seasonal variation characteristics. As shown in Figure 2, wind power output has a certain seasonal variation characteristic. Wind power output is relatively large in autumn and relatively small

in summer, and the difference is relatively obvious. At the same time, the output curve of each season is quite different from the typical solar output curve, and the typical solar output curve cannot reflect well the characteristics of the output variation in each season.

(4) Time series characteristics and similarity. It can be seen from the continuous multi-day variation curve and typical sunrise curve that the wind power output has obvious time series characteristics and similarity. The output of wind power is large at night and small at daytime, which has good peak regulation characteristics. The sunrise curve in continuous time has certain similarity, which means that it can reduce the scene effectively.

From the above analysis, it can be seen that the wind power output has obvious time series characteristics, and the wind power output has obvious differences with different seasons and different periods of the day. The typical daily method cannot adequately express all the information contained in the annual output curve of wind power. At the same time, its contribution has some similarities, which means that it can reduce the necessary scene.

### 2.2.2. Characteristic Analysis of Photovoltaic Output Scene

According to the uncertain modeling results of illumination intensity in the previous section, the corresponding photovoltaic output curve can be obtained. Illumination intensity data can be obtained from the Data Sharing Center of the National Meteorological Administration of China. Figure 5 is the annual photovoltaic output variation curve and its fitting curve of a region. Figure 6 is the photovoltaic output curve of different seasons. Figure 7 is the continuous multi-day photovoltaic output variation curve. Figure 8 is the typical solar output variation curve.

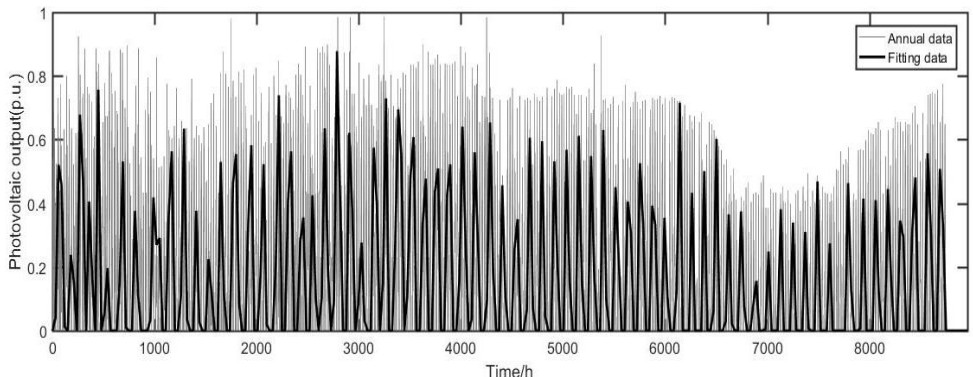

**Figure 5.** Annual wind power output curve.

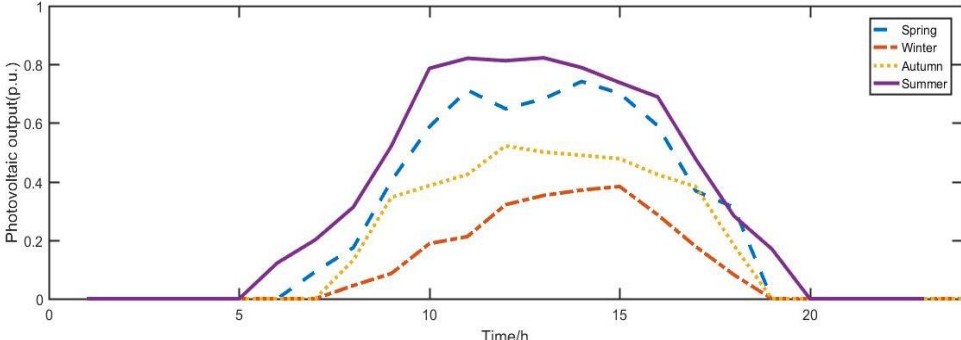

**Figure 6.** Photovoltaic output curve in different seasons.

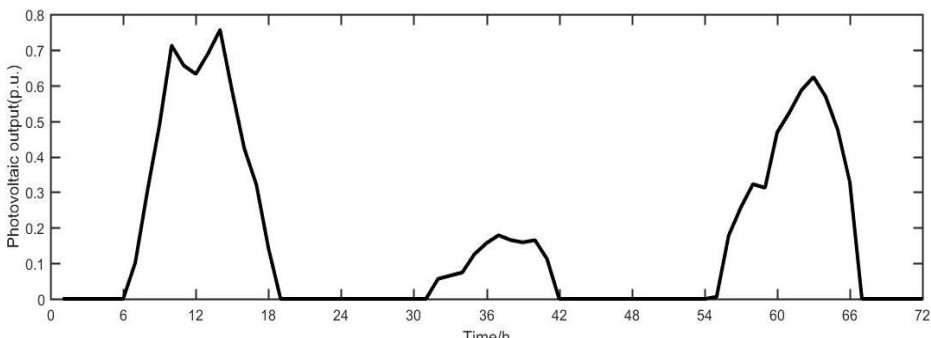

**Figure 7.** Continuous multi-day photovoltaic output curve.

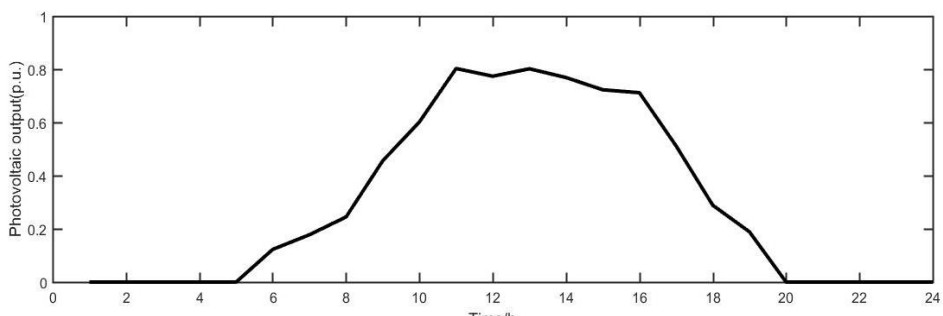

**Figure 8.** Typical photovoltaic output curve.

The characteristics of the above photovoltaic output curves are analyzed and summarized as follows:

(1)　Periodicity. From the solar photovoltaic output curve, it can be clearly observed that the photovoltaic output has obvious periodicity, increasing from the morning until about noon to reach the maximum photovoltaic output, and declining in the afternoon. This periodicity is evident in all curves.

(2)　Seasonal variation characteristics. Photovoltaic output is mainly affected by light intensity. Generally speaking, the photovoltaic output is higher in summer than in winter because of the highest illumination intensity.

(3)　Time series characteristics and similarity. From the continuous multi-day curve and typical daily curve, it can be seen that the photovoltaic output changes obviously with time. On sunny days, the continuous multi-day curves sampled randomly have obvious similarities. This also means that the effort scenario can be reduced.

From the above analysis, it can be seen that the photovoltaic output has obvious time-series characteristics, and the photovoltaic output has obvious differences with different seasons and intra-day periods. A typical daily method cannot adequately express all the information contained in the annual output curve of photovoltaic. At the same time, its contribution also has certain similarity, which means that it can reduce the scene.

## 3. Scene Reduction Method Based on Improved Clustering Algorithm

Cluster analysis is a common method for scene analysis in the DG planning process. Cluster analysis groups the same or similar scenarios in DG and load output scenarios, and obtains the classes of similar elements. Clustering algorithms have been widely used in data analysis. For different sets, different classes are needed, so the clustering algorithms have been improved from the corresponding aspects in the specific application at this stage. In this section, we will introduce the common clustering algorithms. In view of the shortcomings of clustering algorithms and the set of scenarios used in this paper, we propose a kind of improved clustering algorithm.

### 3.1. Improved Clustering Algorithm

According to the scenario characteristics of two kinds of DGs [24], it can be seen that the number of scenario sets in the whole year is large and has certain similarities. Through the introduction of related scene reduction methods, this paper chooses a clustering algorithm to reduce the annual scene set. Intra-class similarity and inter-class difference are the criteria for evaluating clustering algorithm. In order to test the clustering results effectively, this paper chooses BWP index to test the clustering results to judge the reliability of clustering scenarios. This index can also give the optimal number of clustering that traditional clustering algorithm cannot provide.

Let $k = \{X, R\}$ be the clustering space, $X = \{x_1.x_2, \ldots, x_n\}$, Assuming that n objects are eventually clustered into class c, the minimum distance $b(j, i)$ of the sample $i$ in class $j$ is the minimum average distance from the sample to all other types of samples. The concrete expression is shown in Equation (8):

$$b(j, q) = min_{1 \leq k \leq c, k \neq j} \left( \frac{1}{n_k} \sum_{p=1}^{n_k} \|x_p^{(k)} - x_i^{(j)}\|^2 \right) \tag{8}$$

where $x_i^{(j)}$ is the sample $i$ in class $j$; $x_p^{(k)}$ is the sample p in class $K$; $n_k$ is the number of samples in class $k$; and $\| \|^2$ is the square Euclidean distance.

The intra-class distance $w\,(j, i)$ of the sample $i$ in class $j$ is the average distance from the sample to all other samples in class $j$. The concrete expression is shown in Equation (9):

$$w(i, j) = \frac{1}{n_j - 1} \sum_{q=1, q \neq i}^{n_j} \|x_q^{(j)} - x_i^{(j)}\|^2 \tag{9}$$

where $x_q^{(j)}$ is the sample $q$ in class $j$, and $q \neq i$, $n_j$ is the number of samples in class $j$.

$baw(j, i)$ is the sum of the minimum class-to-class distance and the intra-class distance of the sample:

$$baw(j, i) = b(j, i) + w(j, i) \tag{10}$$

$bsw(j, i)$ is the difference between the minimum class-to-class distance and the intra-class distance of the sample:

$$bsw(j, i) = b(j, i) - w(j, i) \tag{11}$$

The index $BWP(j, i)$ of the sample $i$ in class $j$ is the ratio of the clustering distance to the clustering distance of the sample:

$$\text{BWP}(j, i) = \frac{bsw(j, i)}{baw(j, i)} = \frac{b(j, i) - w(j, i)}{b(j, i) + w(j, i)} \tag{12}$$

According to the definition of the BWP index, the bigger the value of BWP index is, the better the clustering result is. The average value of BWP index can reflect the quality of clustering results. When the average value of BWP index is the largest, $k$ is the optimal clustering number. $avgBWP(k)$ is used to represent the average value of BWP indices of all samples when data set $D$ is clustered into $k$ class, and $k_{opt}$ is used to represent the optimal clustering number:

$$avgBWP(k) = \frac{1}{n} \sum_{j=1}^{k} \sum_{i=1}^{n_i} BWP(j, i) \tag{13}$$

$$k_{opt} = argmax_{2 \leq k \leq n} \{avgBWP(k)\} \tag{14}$$

BWP index is used to improve the maximum and minimum distance k-means algorithm, and the best clustering result is determined according to the BWP value. The improved algorithm steps are as follows:

(1)    Choosing a center according to the maximum and minimum distance criterion described above

(2)    Clustering according to k-means clustering method based on maximum and minimum distance

(3)    Calculate the BWP value of the clustering result and turn to step 2

(4)    Comparing the BWP value of clustering results, the $k$ value of clustering results is the best clustering number when the BWP value is maximum

(5)    Clustering results corresponding to the maximum output BWP value

In the process of clustering research and application, there are usually two problems to be solved. One is how to divide a given data set so as to optimize the result. The other is how to divide the data set into the most suitable categories. Among them, the first problem is solved by a clustering algorithm, and the second problem is clustering validation. Although in some applications, the number of clusters can be estimated by a user's experience and domain knowledge, in general, the number of clusters cannot be known in advance, so it is difficult to determine the optimal number of clusters [25].

According to the different components of clustering validity index, it can also be divided into clustering validity index considering only the geometric structure information of data sets, clustering validity index considering only membership degree, and clustering validity index considering both geometric structure information and membership degree of data sets. Among them, data set geometric structure information refers to information extracted from data partition features, such as compactness, separation, connectivity and overlap. Clustering validity index considering only geometric structure information of data sets can be used not only for hard clustering, but also for validity evaluation of fuzzy clustering. Clustering validity index considering only membership degree or considering both geometric structure information and membership degree of data sets can only be used for validity evaluation of fuzzy clustering [26].

Common indicators for k-means clustering algorithm include DB index, I index, CH index, Xie-Beni index, Dunn index, Sil index and so on. The above indicators are tested by artificial simulated data sets and UCI real data sets, respectively [27].

According to the experimental results, the Xie-Beni index, DB index, Dunn index and Sil index give good results only when evaluating the best clustering number of clustering structure features which are far apart and completely separated, but not for other clustering features. Because of the complexity of clustering structure of real data sets, only CH index and I index have significant effect. Xie-Beni index, DB index, Dunn index and Sil index are only good for data sets with 2 clustering numbers because of their own limitations. Therefore, CH index and I index are two good choices when evaluating the optimal clustering number of k-means algorithm.

It should be pointed out that when the scene reduction method based on improved clustering algorithm is used to reduce the specific scene, the reduced scene with larger BWP value can be selected according to the actual scene reduction requirement rather than the maximum value. Choosing the reduced set corresponding to the high $k$ value can make use of the time series characteristic of retaining the original scene set to a greater extent.

*3.2. Validity Test*

3.2.1. Scene Reduction Process Based on Improved Clustering Algorithm

Scene reduction is the process of classifying and merging objects to be clustered. According to the results of past research and the analysis of scenario characteristics in Section 2, the wind farm scenario set is divided into four scenarios corresponding to spring, summer, autumn and winter. The photovoltaic scenic set is divided into 12 scenarios in spring, summer, autumn, winter and three weather types: cloudy, sunny and rainy. When scene reduction of DG is carried out, scene reduction is carried out with day as the basic unit of clustering. Assuming that the total number of individual scenarios $n(1, 2, \dots, N)$ is $N$. A single scenario has $T$-period scenario data. The data contained in all scenarios can be represented by matrix $N * T$. By improving the clustering algorithm to merge the same kind of scene into $K$ scenes, the reduced scene data can be represented by $K * T$. Typical scenes

obtained after reduction have the same temporal characteristics as the original scenes, so as to ensure the temporal characteristics of the scenes before reduction. Scenario reduction of two types of DGs and loads is carried out by using the above method. This paper takes the wind power generation scenario as an example to test the effectiveness of the improved clustering algorithm proposed.

The scene reduction process based on improved clustering algorithm is shown in Figure 9.

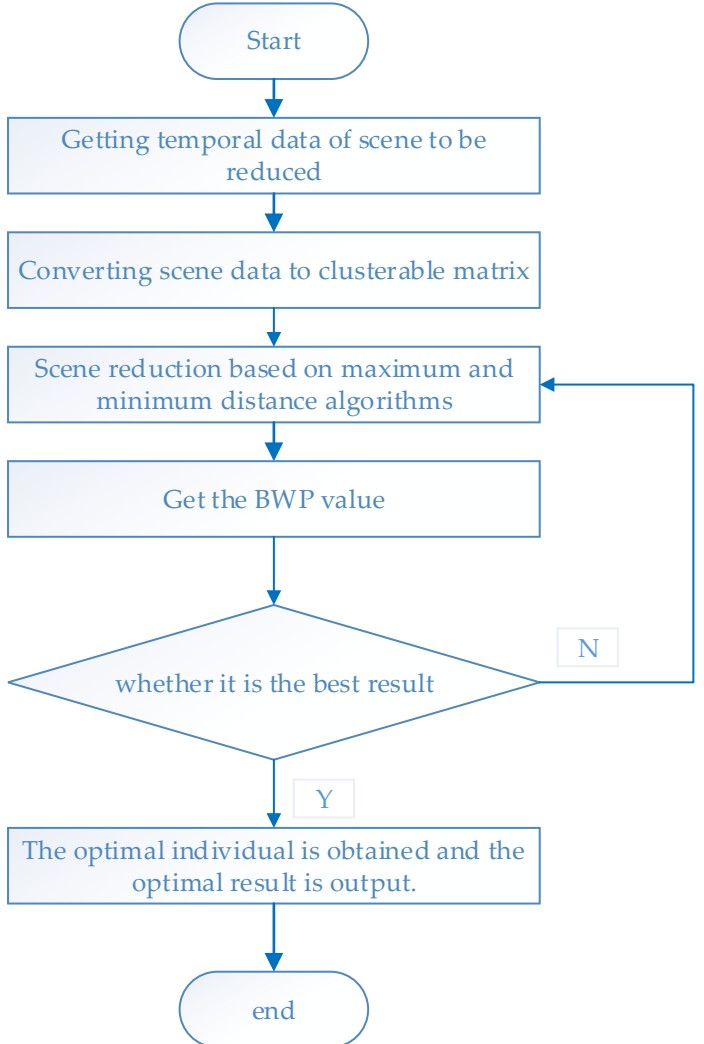

**Figure 9.** Scene Reduction Method of Improved Clustering Algorithms.

### 3.2.2. Validity Test of Wind Power Output

In this paper, wind power generation scenarios are taken as an example to verify the effectiveness of the reduced scenarios. According to the four seasons of spring, summer, autumn and winter, all scenes are divided. The improved clustering algorithm proposed in this paper is used to reduce the partitioned scenes, and the BWP value of clustering results is calculated to select the optimal result. Figure 10 shows the change curve of BWP value with $k$ value after clustering.

According to the change curve of BWP value, the BWP value is the largest in spring, autumn and winter scenarios when $k = 2$, and in summer, when $k = 3$, the BWP value is the largest. But in order to reflect the temporal characteristics of the original scene to the greatest extent, this paper chooses the case where the k value is relatively large and the number of scenes is relatively large. Taking spring as an example, when $k = 2$ and $k = 5$, the k value is larger. Choosing $k = 5$ here, the scenarios to be reduced are divided into five categories. The output curves of these five scenarios are given below, as shown in Figure 11.

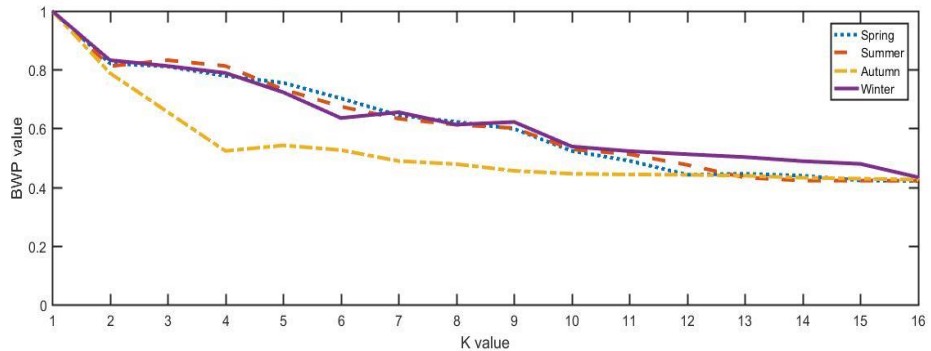

**Figure 10.** BWP Value Change Curve.

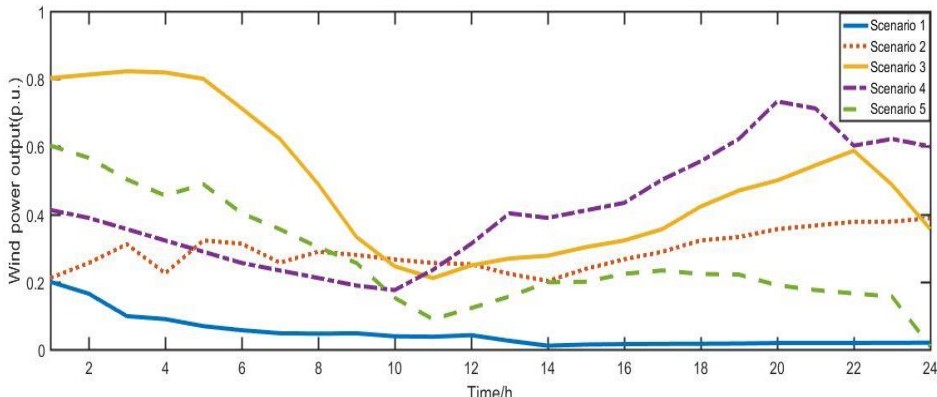

**Figure 11.** Spring Output Curve at $k = 5$.

According to the law of large numbers, the corresponding probability of each scene at $k = 5$ is shown in Table 1.

**Table 1.** The probability of typical temperatures throughout one year.

| Season | Scene Reduction | | | | |
|---|---|---|---|---|---|
| | 1 | 2 | 3 | 4 | 5 |
| Spring | 0.57 | 0.21 | 0.07 | 0.12 | 0.03 |

In order to reflect the relationship between the reduced scene and the original scene more clearly, scene No. 5 is selected, and two scenes are randomly selected from the reduced scene set for comparative analysis, as shown in Figure 12.

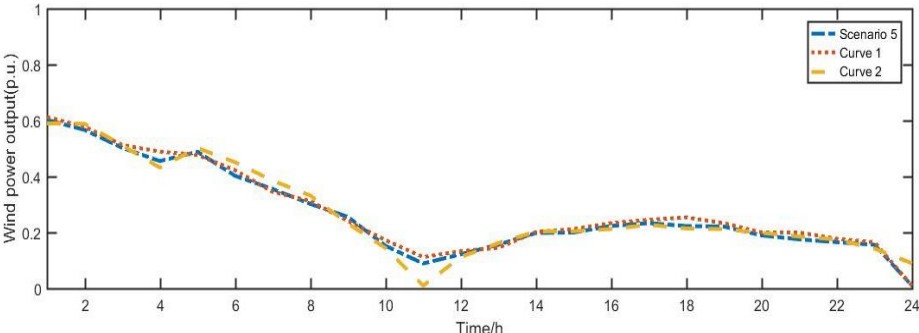

**Figure 12.** Scene reduction contrast graph.

According to the reduced output, nearly half of the wind power output in Figure 11 is as shown in Scenario 1. In the other scenarios, the wind power output shows obvious peak reversal characteristics. In the second section of this paper, the intra-day variation characteristics of wind power output obtained from scenario characteristics analysis are better reflected in different reduced scenarios. Through the verification of BWP value, and from Figure 12, we can see that the reduced scene obtained by the improved clustering algorithm in this paper has better coincidence with the original scene and can better reflect the temporal characteristics of the original scene.

### 3.2.3. Scene Reduction for Two Kinds of Intermittent DG

The scene reduction method based on improved clustering algorithm is used to reduce the output curves of two kinds of DGs, and the validity test is carried out. The results of scene reduction are given directly here:

$$\delta_i(t) = P_{wav}(t)/P_T \tag{15}$$

After reducing and normalizing the wind generator scenes divided by seasons, the typical daily scenes are shown in Figure 13.

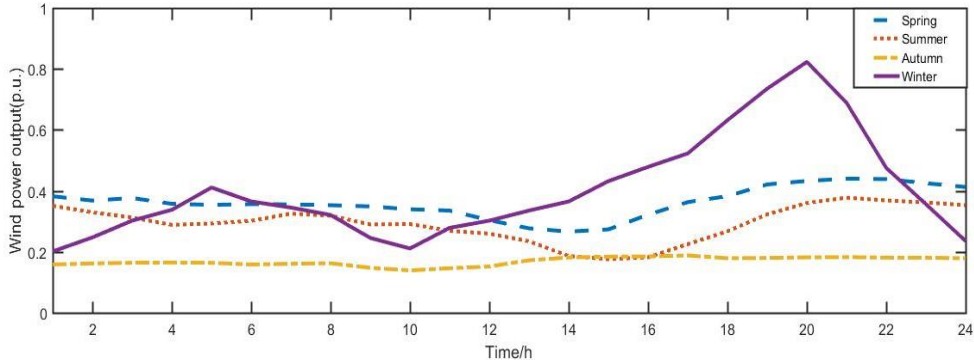

**Figure 13.** Wind power curve after scene reduction.

After reducing and normalizing the photovoltaic power generation scenarios divided by season and weather, the typical daily scenarios are shown in Figures 14–17, respectively.

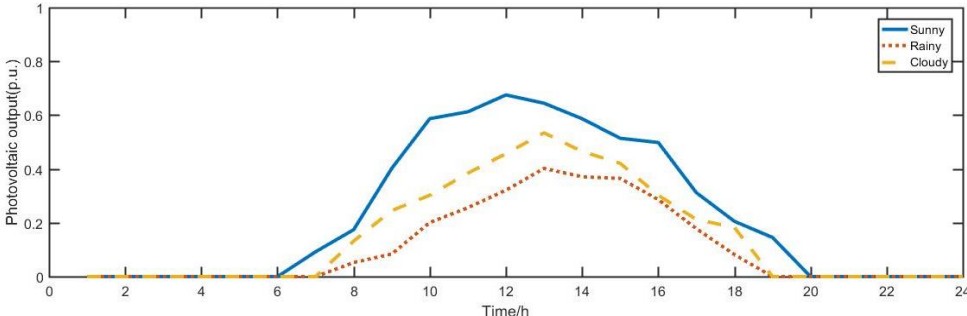

**Figure 14.** Photovoltaic curve after scene reduction in spring.

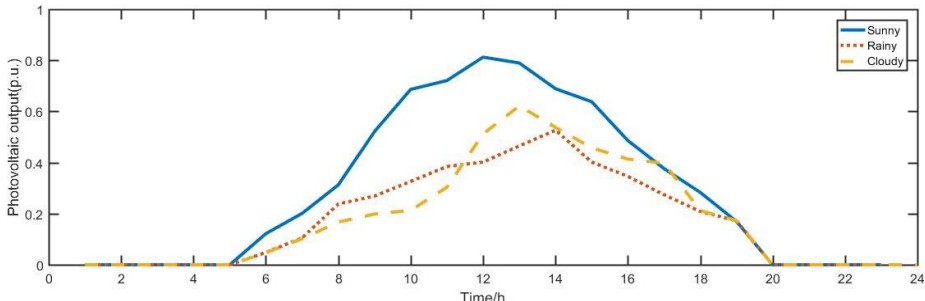

**Figure 15.** Photovoltaic curve after scene reduction in summer.

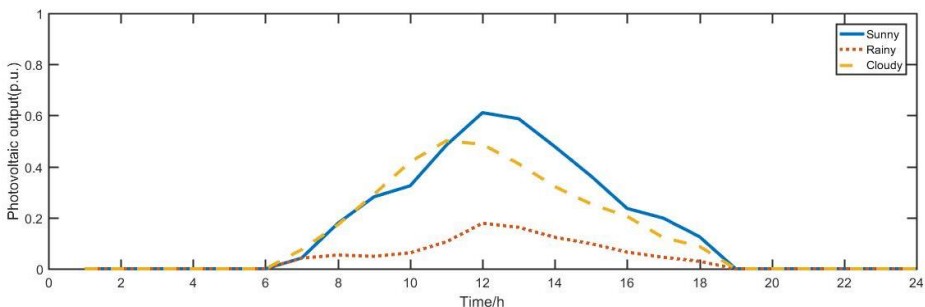

**Figure 16.** Photovoltaic curve after scene reduction in autumn.

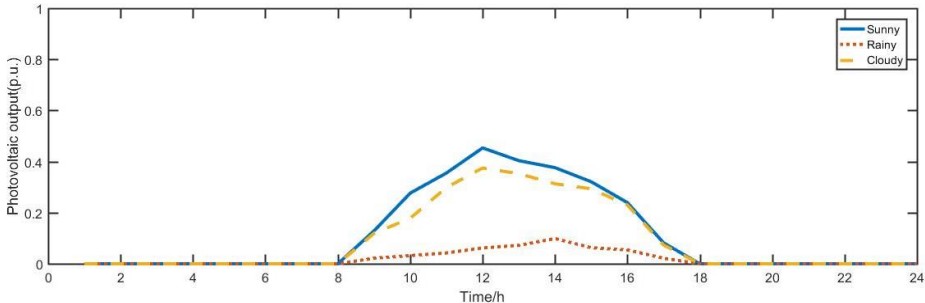

**Figure 17.** Photovoltaic curve after scene reduction in winter.

## 4. Examples and Analysis

In the existing DG planning model, the influence of active management mode on the distribution network planning is generally considered. However, the ADN active control is generally not considered in the model to maintain the safe and stable operation of islands under the failure state. This paper establishes an upper-level programming model aiming at minimizing the annual comprehensive cost. In the upper-level planning, ADN operation strategy in fault scenario is considered, and the comprehensive safety index is introduced and converted into upper-level constraints. To minimize the amount of active power cut-off, this paper adopts the following three kinds of active management measures [24]:

(1)  Distributed generator output control
(2)  Switching of reactive power compensation
(3)  Adjustment of on load transformer

*4.1. Upper Level Programming Model and Lower Level Programming Model*

The upper-level programming model considers DG layout and installation capacity planning. The objective function is to minimize the annual life cycle investment cost:

$$minC_1 = C_I + C_{OM} + C_P + C_{AM} + C_L \tag{16}$$

The specific expressions of each cost are as follows:

(1) DG Equivalent Investment Annual Cost

$$C_I = \left( \sum_{i=1}^{N_{bus}} C_{WTG,i} S_{WTG,i} + \sum_{i=1}^{N_{bus}} C_{PVG,i} S_{PVG,i} \right) \frac{r(1+r)^y}{(1+r)^y - 1} \tag{17}$$

where $N_{bus}$ is the number of nodes in the distribution network, $r$ is the discount rate, $y$ is the life span of DG for 20 years, $C_{WTG,i}$ and $C_{PVG,i}$ are fixed investment costs of unit capacity of wind power and PV installed at the $i$ node respectively, $S_{WTG,i}$ and $S_{PVG,i}$ are the rated capacity of wind power and PV installed at the $i$ node respectively.

(2) DG annual operation and maintenance costs

$$C_{OM} = \sum_{n=1}^{12} p_n \times 365 \left( \sum_{t=1}^{24} \left( \sum_{i=1}^{N_{bus}} C_{WTG,i} E_{WTG,in}(t) + \sum_{i=1}^{N_{bus}} C_{PVG,i} E_{PVG,in}(t) \right) \right) \tag{18}$$

where $P_n$ is the scenario probability of the $n$ scenario, $C_{WTG,i}$ and $C_{PVG,i}$ are the operation and maintenance costs of the wind power and the photovoltaic unit electricity received by the $i$ node, $E_{WTG,in}(t)$ and $E_{PVG,in}(t)$ are the wind power received by the $i$ node and the photovoltaic unit electricity generated during the $t$ period of the $n$ typical day, respectively.

(3) Annual Electricity Purchase Cost

$$C_P = \sum_{n=1}^{12} P_n \times 365 \left( \sum_{t=1}^{24} E_{nt} P_t \right) \tag{19}$$

where $E_{nt}$ is the $t$ time of $n$ typical days to buy electricity from a higher power grid. $P_t$ is the unit cost of operators purchasing electricity from a higher power grid.

(4) DG annual active management cost

$$C_{AM} = \sum_{n=1}^{12} p_n \times 365 \left( \sum_{t=1}^{24} \left( \sum_{i=1}^{N_{bus}} C_{AWTG,i} E_{WTG,in}(t) + \sum_{i=1}^{N_{bus}} C_{APVG,i} E_{PVG,in}(t) \right) \right) \tag{20}$$

where $C_{AWTG,i}$ and $C_{APVG,i}$ are the active management costs of the wind power and the photovoltaic at the $i$ node respectively.

(5) Network Loss Cost

$$C_L = \sum_{n=1}^{12} P_n \times 365 \left( \sum_{t=1}^{24} Q_{ntL} P_{ntL} \right) \tag{21}$$

where $Q_{ntL}$ is the net loss of the $n$ typical day $t$ period, $P_{ntL}$ is the unit network loss cost.

The constraints are:

(1) DG Installation Capacity Limitation

$$\begin{aligned} 0 \le R_{WTGi} \le R_{WTGmax} \\ 0 \le R_{PVGi} \le R_{PVGmax} \end{aligned} \tag{22}$$

where $R_{WTGi}$ and $R_{PVGi}$ are the wind capacity and PV capacity node i respectively. $R_{WTGmax}$ and $R_{PVGmax}$ correspond to the maximum access capacity of DG, respectively.

(2) Capacity Limitation of DG Total Installation

$$\sum_{i=1}^{N_{WTG}} R_{WTGi} + \sum_{i=1}^{N_{PVG}} R_{PVGi} \leq R_{DGmax} \tag{23}$$

where $R_{DGmax}$ is the maximum installed capacity. The determination of maximum access capacity needs to consider different adjustment strategies.

(3) Constraints of Comprehensive Safety Indicators

$$C_{csi} = \frac{1}{2}\left(\frac{1}{NTS}\sum_{n=1}^{N}\sum_{t=1}^{T}\sum_{s=1}^{S} C_{n,t,s} + min\{C_{n,t,s}\}\right) \tag{24}$$

where $C_{n,t,s}$ is the index of safe power supply rate of branch $s$ of the $n$ scenario at $t$ period.

$$C_{n,t,s} = 1 - \frac{\sum_{t=1}^{t_T}\sum_{y\in\varphi_{ntf}}\gamma_y S_{n,t,y}\Delta T_{n,t,y}}{\sum_{t=1}^{t_T}\sum_{i\in\varphi_{ntl}}\gamma_y S_{n,t,y}\Delta T_{n,t}} \tag{25}$$

where $t_T$ is the period of system failure elimination. $\varphi_{ntf}$ is the $n$ scenario of $t$ period outage load set. $\gamma_y$ is the grade factor of Class $y$ load. $S_{n,t,y}$ is the Capacity of Class $y$ Load. $\Delta D_{n,t,y}$ is the outage time of $y$-load in the nth scenario at t period. $\varphi_{ntl}$ is t period load set for the n scenario.

The lower level planning model mainly considers the operation constraints related to the operation of distribution network. At this stage, DG access to power grid costs higher. In order to maximize the utilization of DG, the lower level objective is to minimize the amount of active power cut-off of DG, and its expression is as follows:

$$minC_2 = \sum_{n=1}^{12} P_n \times 365\left(\sum_{t=1}^{24} P_{cnt}\right) \tag{26}$$

The constraints are:

(1) Node Power Balance Constraints

$$\begin{aligned} P_{ci,i,t,n} + P_{co,i,t,n} - P_{WTG,i,t,n} - P_{PVG,i,t,n} = \\ U_{i,t,n}\sum_{j=1}^{N_{bus}} U_{j,t,n}(G_{ij}cos\theta_{t,n,ij} + B_{ij}sin\theta_{t,n,ij}) \end{aligned} \tag{27}$$

$$\begin{aligned} Q_{ci,i,t,n} + Q_{co,i,t,n} - Q_{WTG,i,t,n} - Q_{PVG,i,t,n} - Q_{c,i,t,n} = \\ U_{i,t,n}\sum_{j=1}^{N_{bus}} U_{j,t,n}(G_{ij}sin\theta_{t,n,ij} - B_{ij}cos\theta_{t,n,ij}) \end{aligned} \tag{28}$$

where $P_{WTG,i,t,n}$ and $P_{PVG,i,t,n}$ are the active output of the t time of the n scenario, respectively. $P_{ci,i,t,n}$ and $P_{co,i,t,n}$ are the active power of residential and commercial loads at the first time of $t$ in the first $n$ scenario, respectively. $Q_{WTG,i,t,n}$ and $Q_{PVG,i,t,n}$ are the reactive power of DG at the $t$ time of the n scenario, respectively. $Q_{ci,i,t,n}$ and $Q_{co,i,t,n}$ are the reactive power of resident load reactive power and commercial load at the t time of the n scenario, respectively, supplied by the reactive power compensation device. $U_{i,t,n}$ and $U_{j,t,n}$ are the voltage amplitude of node $i$ and the voltage amplitude of node $j$ at the $t$ time node of the $n$ scenario, respectively. $\theta_{t,n,ij}$ is the phase difference between node $i$ and node $j$ of t in the $n$ scenario, respectively.

(2) Node Power Balance Constraints

$$U_{imin} \leq U_i \leq U_{imax} \tag{29}$$

where $U_i$ is node voltage. $U_{imin}$ and $U_{imax}$ are the minimum voltage values and maximum voltage values allowed by node *I*, respectively.

(3) Branch power constraints

$$S_i \leq S_{imax} \tag{30}$$

where $S_i$ is the apparent power of branch *L*. $S_{imax}$ is the limit of branch transmission capacity.

(4) DG output control constraints

$$P_{imin} \leq P_i \leq P_{imax} \tag{31}$$

where $P_{imin}$ and $P_{imax}$ are the minimum active power output of node *i* and the maximum active power output of distributed generation respectively.

(5) Constraints of reactive power compensation

$$Q_{imin} \leq Q_i \leq Q_{imax} \tag{32}$$

where $Q_{imin}$ and $Q_{imax}$ are the minimum value of reactive power compensation device of node *I* and the maximum value of the reactive power compensation device.

(6) Regulation constraints of on-load tap-changer

$$T_{imin} \leq T_i \leq T_{imax} \tag{33}$$

where $T_i$ is the tap position of transformer *i*. $T_{imin}$ and $T_{imax}$ are the tap values of transformer *i* and the maximum tap value of *i*, respectively.

### 4.2. Bi-Level Programming Model Solving Algorithms

The solution of mixed non-integer programming problem is a common problem in the process of optimal allocation of distributed power supply, and its essence is a NP-hard problem. At present, heuristic algorithm and deterministic algorithm are the main solving methods. Particle swarm optimization (PSO), genetic algorithm (GA) and related improved algorithms are widely used in DG programming. In this paper, a cuckoo search algorithm is used to solve the upper model, and the lower model is solved by the original dual interior point method.

#### 4.2.1. Cuckoo Search Algorithms

The cuckoo search algorithm was first proposed in 2009 by Yang and Deb. The CS algorithm can efficiently search the optimal solution of the problem by simulating the parasitic brooding of some species of cuckoo. At the same time, CS also uses the relevant Levy flight search mechanism.

In the process of cuckoo reproduction, the nest location of cuckoo's offspring is uncertain. In the process of simulating its search for bird's nest, three principles need to be recognized:

(1) Cuckoos lay only one egg at a time of reproduction, and then they choose their nests arbitrarily for hatching and rearing.
(2) The most suitable nest will be extended to the next generation of reproduction in a randomly selected set of options.
(3) The total number of nests available *N* is a fixed value, and the probability that the original owner of the nest has $P_a \in [0, 1]$ can identify a non-self-laid bird's egg.

Based on these three principles, the path and location of cuckoo nest selection are determined by Equation (34):

$$x_i^{e+1} = x_i^e + \alpha * L(\lambda), i = 1, 2, \ldots, n \tag{34}$$

where, $x_i^e$ is the position of the i nest in the selection of the e generation; * is point-to-point multiplication; $\alpha$ is the step size control in the process of choosing nests for cuckoos, which obeys the normal distribution; $L(\lambda)$ is the path through which Levy searches bird's nest arbitrarily, and $L(d, \lambda) \sim s^{-\lambda}(1 < \lambda \leq 3)$, d is the random step obtained by Levy's flight.

### 4.2.2. Bi-Level Programming Model Solving Process

The detailed flow chart of solving multi-objective bi-level programming model by CS algorithm and PDIPM method is shown in Figure 18. Among them, G is the number of iterations.

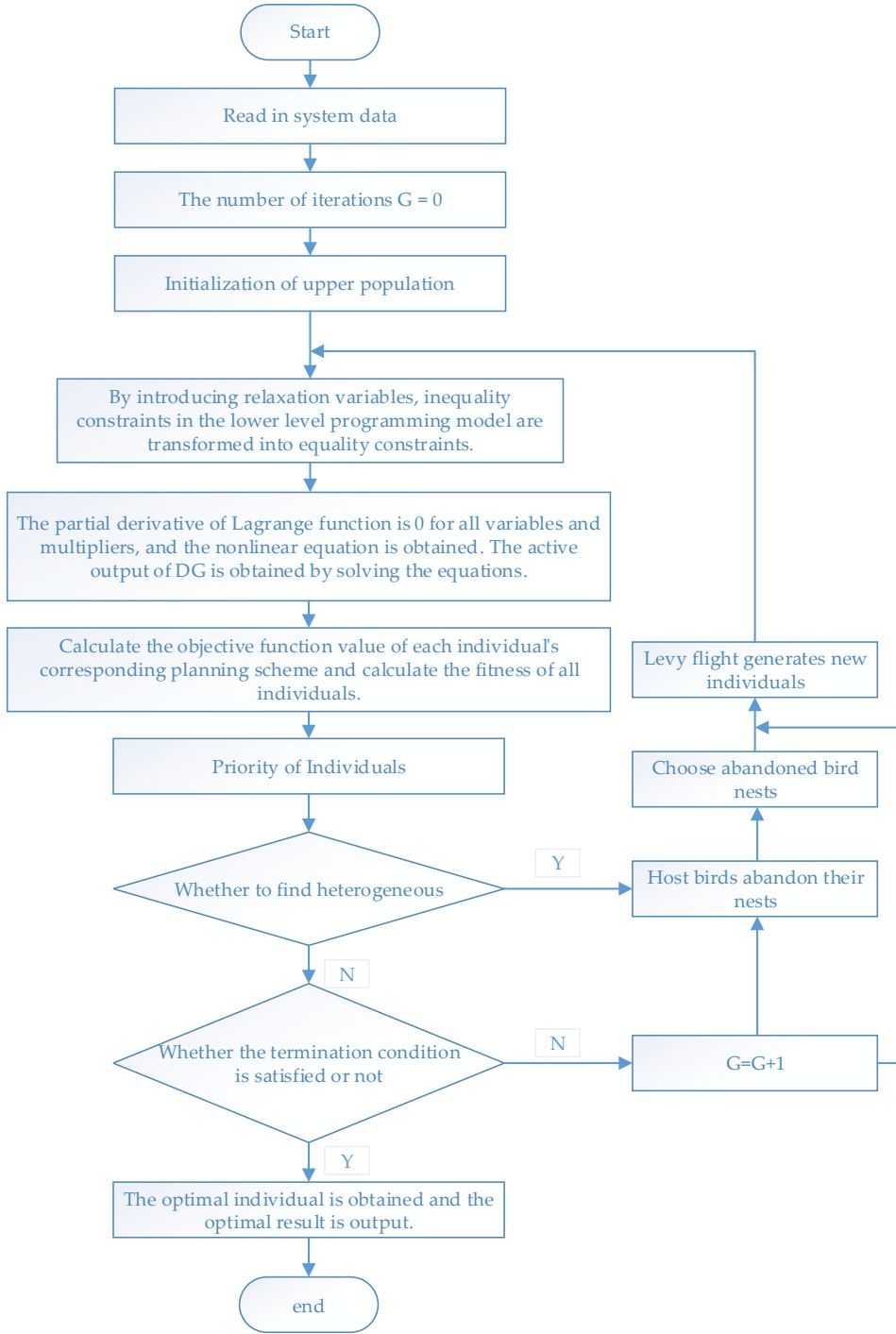

**Figure 18.** Bi-level planning model solving process.

*4.3. Introduction of Examples*

The proposal is verified on using the IEEE33 node system which is shown in Figure 19 (Figure 19). The system voltage is 12.66 kV, total active load is 3.715 MW, total reactive load is 2.300 MW, Weibull distribution parameter k = 2.30, C = 8.92, wind power access cost 6500 yuan/kW, operation and maintenance cost 0.3 yuan/kW·h, environmental protection subsidy 0.1 yuan/kW·h, rated lighting intensity of photovoltaic generator is 1 kW/m², shape parameter B of beta distribution is 0.85, photovoltaic access The cost is 10,000 yuan/kW, the operation and maintenance cost is 0.2 yuan/kW·h, the rated capacity of a single distributed power supply is 125 kW, the equipment life is 20 years, and the discount rate is 0.1. The repair time of N-1 fault is 4 h, and the comprehensive safety index value is set to 0.5. Wind power installation nodes are 5, 7, 11, 12. The photovoltaic installation nodes are 20 and 23.

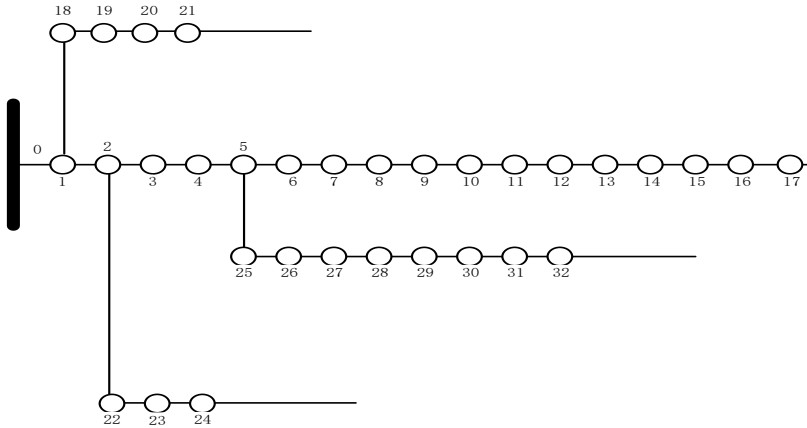

**Figure 19.** IEEE-33 node distribution system.

On the premise of considering both comprehensive security indicators and active management mode, the economic and technical comparison of DG optimal allocation under the annual time series scenarios, typical day scenarios and reduced scenarios is made. Full-time scenarios are selected as benchmarks to test the planning schemes under the other two scenarios. The annual average of each period is selected as the data of typical day scenes. A summary of the three plans is shown in Table 2.

**Table 2.** Comparison of three kinds of scene set planning schemes.

| Pattern | DG | Installation Node | Installation Capacity/kW | Annual Life Cycle Investment Cost/¥ | Active Power Excision MW · h | Computing Time/s |
|---------|-----|------------------|-------------------------|------------------------------------|----------------------------|------------------|
| Annual Sequence Scene | WG | 5 | 125 | 5,123,600 | 48.26 | 1803 |
| | | 7 | 125 | | | |
| | | 11 | 375 | | | |
| | | 12 | 250 | | | |
| | PV | 20 | 125 | | | |
| | | 23 | 250 | | | |
| Typical Day Scene | WG | 5 | 125 | 6,145,600 | 31.48 | 8 |
| | | 7 | 125 | | | |
| | | 11 | 250 | | | |
| | | 12 | 125 | | | |
| | PV | 20 | 125 | | | |
| | | 23 | 125 | | | |
| Scene Reduction | WG | 5 | 125 | 4,547,400 | 45.77 | 60 |
| | | 7 | 125 | | | |
| | | 11 | 250 | | | |
| | | 12 | 250 | | | |
| | PV | 20 | 125 | | | |
| | | 23 | 250 | | | |

## 5. Discussion

From the comparison results of three scenario planning schemes, it can be concluded that:

Most directly, when using reduced scene set for DG planning, the computational time is reduced from about 1803 s to only about 60 s, and the computational efficiency has been significantly improved. Although the time spent in DG planning using reduced scene sets is slightly longer than that using typical day scenes, the calculation accuracy is higher. In this paper, DG with fixed capacity is used to access the corresponding nodes. Under this assumption, the DG access capacity in reduced scenarios is the same as that in annual sequential scenarios, and the typical daily scenario with average value is smaller than the other two scenarios. Compared with the typical Japanese method, the annual comprehensive cost and the amount of effective removal of reduced scenes are closer to the annual time series scenes. In DG planning process, the access capacity and location of the reduced scene are close to that of the year-round sequential scene. In summary, the scene reduction method based on improved clustering algorithm proposed in this paper has better retention effect for the original scene time series data, and the economic and technical indicators can basically accurately reflect the sequence status of the scene before reduction.

## 6. Conclusions

The main purpose of this paper is to propose a scene reduction method based on improved clustering algorithm. The main problem solved in this paper is how to use the multi-scenario analysis method to analyze the output characteristics of distributed power supply, taking into account the calculation efficiency and accuracy. In this paper, the validity of the related algorithms is verified, and the scene reduction method is compared with the common methods by using ieee-33 node system. The details are as follows:

(1)  Uncertainty modeling of two kinds of distributed generators is carried out. The scene output characteristics of DG are analyzed. The analysis shows that both DGs have obvious uncertainties and time series characteristics, therefore, the typical scenes composed of the average method or the maximum sunrise of peak-valley difference cannot effectively reflect the characteristics of DG. At the same time, the output of distributed generation has some inherent similarities, which means that it can be effectively reduced.

(2)  To overcome the shortcomings of large-scale scene clustering algorithms, a scene reduction method based on an improved clustering algorithm is proposed, and its validity is tested. The test results show that the improved clustering algorithm can effectively retain the characteristics of the original output scenario.

(3)  Through a specific example, the scene reduction method is verified by using IEEE33 node system. The three scenario sets show that the proposed scenario reduction method is feasible. The scenario constructed by this method has good preservation effect on the original scenario, and can take into account both the computational efficiency and the computational accuracy in the process of distributed generation optimal configuration.

**Author Contributions:** All authors contributed to the research. Data curation, S.L., Y.G. and Z.S.; Formal analysis, S.L.; Project administration, J.L.; Writing—original draft, S.L.; Writing—review & editing, S.L.

**Funding:** This work is supported by the National Natural Science Foundation of China (Grant No. 51477099); Scientific Research Projects of Shanghai Science and Technology Commission (Grant No. 17DZ1201200).

**Conflicts of Interest:** The authors declare no conflict of interest.

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
