# Peer review of "A Typical Distributed Generation Scenario Reduction Method Based on an Improved Clustering Algorithm"

_applsci, doi:10.3390/app9204262_

Round 1
Reviewer 1 Report
Authors are suggested to address the following comments.
Abstract, the abbreviation for distributed generation should be defined in first sentence but not in line 15. Title should be matched with the main theme of the paper. For instance, distributed generation and improved clustering algorithm should be included. Keywords should match with the main theme of the paper. Distributed generation must be included. Introduction, paragraph 1, reference is missing for the statement “According to the current consumption rate, the oil stock can only be used for 40 years, the natural gas village stock can only be used for 60 years, and the coal stock can be used for 200 years.” Introduction, paragraph 1, what is the reason for the claim of “The energy crisis caused by over-exploitation of non-renewable energy restricts the rapid development of economy.”? Introduction, authors mentioned ” This paper mainly introduces the common clustering algorithms in scene reduction, and then proposes a scene reduction method based on an improved clustering algorithm.”. Please clarify if authors are adopting traditional clustering algorithms or improved clustering algorithm (i.e. with customization). Relevant and updated related works (mainly cover 2015-2019 papers) are missing. Introduction, it is suggested to summarize the contributions in point-form. Does this paper incomplete? It starts with Section 1 which is followed up Section 2.1.2. Equations (1)-(3) should be moved to methodology rather than presenting in introduction. Equation (3), P_WTG has not been defined. Equations (1)-(7), please provide details and justifications for the parameter settings. Authors mentioned “Figure 1 is the annual wind power output variation curve of a certain area. In order to observe its variation more intuitively, the fitting curve is made. Figure 2 is the average wind power output variation curve in different seasons, Figure 3 is the continuous multi-day wind power output variation curve drawn randomly, and Figure 4 is the typical wind power output curve”, there is no proper written description to explain the source, formulation, etc. of the curves. Similarly, please apply the same for Figures 5-8 which authors mentioned “Figure 5 is the annual photovoltaic output variation curve and its fitting curve of a region. Figure 6 is the photovoltaic output curve of different seasons. Figure 7 is the continuous multi-day photovoltaic output variation curve. Figure 8 is the typical solar output variation curve.”. Figures 1-8 are not considered as technical contribution of this paper. What are the purposes of presenting these contents? Section 3.1, line 203, authors mentioned “…such as equation 8:”, does it mean there are many more expressions? Please comment on the feasibility of calculating optimal number of clusters by CH index, DB index and SH index. Please cite the following article “Appliance signature identification solution using K-means clustering”. Caption of figure 9 should be revised. It should be about the improved clustering algorithm. Please be consistent in using “k” or “K”. Please specify the scenarios in Figure 11. Table 1, how about the results for summer, autumn and winter? Figure 12, why only scenario 5 is being considered? The contribution is not clear.
Author Response
Response to Reviewer 1 Comments
Point 1: Abstract, the abbreviation for distributed generation should be defined in first sentence but not in line 15.
Response 1: Necessary changes have been made to the corresponding content.
Point 2: Title should be matched with the main theme of the paper. For instance, distributed generation and improved clustering algorithm should be included. Keywords should match with the main theme of the paper. Distributed generation must be included
Response 2: The title and key words have been revised.
Point 3: Introduction, paragraph 1, reference is missing for the statement “According to the current consumption rate, the oil stock can only be used for 40 years, the natural gas village stock can only be used for 60 years, and the coal stock can be used for 200 years.” Introduction, paragraph 1, what is the reason for the claim of “The energy crisis caused by over-exploitation of non-renewable energy restricts the rapid development of economy.”? Introduction, authors mentioned ” This paper mainly introduces the common clustering algorithms in scene reduction, and then proposes a scene reduction method based on an improved clustering algorithm.”. Please clarify if authors are adopting traditional clustering algorithms or improved clustering algorithm (i.e. with customization). Relevant and updated related works (mainly cover 2015-2019 papers) are missing. Introduction, it is suggested to summarize the contributions in point-form.
Response 3: A large number of amendments have been made to the introduction, and some inaccurate contents have been deleted. This paper extends the references and adds new contents so that readers can understand the contents of this article more accurately and easily. You can refer to the new uploaded document for the corresponding content of the modification.
Point 4: Does this paper incomplete? It starts with Section 1 which is followed up Section 2.1.2. Equations (1)-(3) should be moved to methodology rather than presenting in introduction.
Response 4: The missing section is mainly due to typographical errors. This revision adds a new content, and according to the new text, new arrangements have been made for the whole article section. You can refer to the new uploaded document for the corresponding content of the modification.
Point 5: Equation (3), P_WTG has not been defined. Equations (1)-(7), please provide details and justifications for the parameter settings. Authors mentioned “Figure 1 is the annual wind power output variation curve of a certain area. In order to observe its variation more intuitively, the fitting curve is made. Figure 2 is the average wind power output variation curve in different seasons, Figure 3 is the continuous multi-day wind power output variation curve drawn randomly, and Figure 4 is the typical wind power output curve”, there is no proper written description to explain the source, formulation, etc. of the curves. Similarly, please apply the same for Figures 5-8 which authors mentioned “Figure 5 is the annual photovoltaic output variation curve and its fitting curve of a region. Figure 6 is the photovoltaic output curve of different seasons. Figure 7 is the continuous multi-day photovoltaic output variation curve. Figure 8 is the typical solar output variation curve.”
Response 5: Detailed definitions are given for each parameter in the formula. Specific parameter settings, data sources and equipment models are added to the validity test and case analysis respectively. You can refer to the new uploaded document for the corresponding content of the modification.
Point 6: Figures 1-8 are not considered as technical contribution of this paper. What are the purposes of presenting these contents?
Response 6: The main work of this paper is to reduce the output curve of distributed generation. Before reducing the output curve, necessary analysis must be made to judge the possibility of scene reduction. Figure 1-8 is exactly what is going on.
Point 7: Section 3.1, line 203, authors mentioned “…such as equation 8:”, does it mean there are many more expressions?
Response 7: I'm sorry, but this is a mistake in language expression.
Point 8: Please comment on the feasibility of calculating optimal number of clusters by CH index, DB index and SH index.
Response 8: The clustering number has been studied in relevant literature. https://kns.cnki.net/KCMS/detail/detail.aspx?dbcode=CJFQ&dbname=CJFD2014&filename=XTLL201409027&v=MDE5MzZZNFI4ZVgxTHV4WVM3RGgxVDNxVHJXTTFGckNVUkxPZmJ1UnRGeTdrVkw3S1BUbkhZckc0SDlYTXBvOUg=
You can see it in the manuscript in the link above. If you feel it necessary, I will do some research and discussion in my manuscript in the next revision.
Point 9: Please cite the following article “Appliance signature identification solution using K-means clustering”. Caption of figure 9 should be revised. It should be about the improved clustering algorithm. Please be consistent in using “k” or “K”.
Response 9: Corresponding content has been revised in the new manuscript.
Point 10: Please specify the scenarios in Figure 11. Table 1, how about the results for summer, autumn and winter? Figure 12, why only scenario 5 is being considered? The contribution is not clear.
Response 10: In the process of validity test, only part of the content is tested because the other parts are tested in the same way. In order to show the results of other content better and avoid unnecessary repetition, we directly give the results of scene reduction. In order to test the reduced results, we add a case study in this revision, so as to better demonstrate the methods proposed in this paper.

Reviewer 2 Report
The topic of the paper is interesting and within the scope of the journal. The following changes must be done prior its publication.
The use of English must be improved. Several grammatical and syntax errors must be corrected.
A more representative title of the presented work must be selected.
The Introduction must be extended. The already mentioned papers must be further commented while others must be included in order the current state of the art and the research area to be clear even for unfamiliar readers. The following papers must be included:
Vita V., Alimardan T., Ekonomou L., The impact of distributed generation in the distribution networks’ voltage profile and energy losses, Proceedings of the 9th IEEE European Modelling Symposium on Mathematical Modelling and Computer Simulation, Madrid, Spain, pp. 260-265, 2015.
Vita V., Development of a decision-making algorithm for the optimum size and placement of distributed generation units in distribution networks, Energies, Vol. 10, No. 9, (DOI) 10.3390/en10091433, 2017.
Nieto A., Vita V., Ekonomou L., Mastorakis N.E., Economic analysis of energy storage system integration with a grid connected intermittent power plant, for power quality purposes, WSEAS Transactions on Power Systems, Vol. 11, 2016, pp. 65-71.
Section 2 main title and subsection’s 2.1 title are missing.
The authors should avoid having so many subsections. It is better to have more main sections.
The results of the proposed methodology must be compared with these of other similar methodologies.
A Discussion section must be included that will comment on the produced results.
Conclusions must summarize the work presented in the paper.
Author Response
Response to Reviewer 2 Comments
Point 1: The use of English must be improved. Several grammatical and syntax errors must be corrected.
Response 1: Necessary revisions and checks have been made to the English language of the full text.
Point 2: A more representative title of the presented work must be selected.
Response 2: A more representative title of the presented work has been selected.
Point 3: The Introduction must be extended. The already mentioned papers must be further commented while others must be included in order the current state of the art and the research area to be clear even for unfamiliar readers. The following papers must be included:
Response 3: Necessary modifications have been made to the introduction. The references are expanded and new ones are added so that readers can better understand the content of this article.
Point 4: Section 2 main title and subsection’s 2.1 title are missing. The authors should avoid having so many subsections. It is better to have more main sections. The results of the proposed methodology must be compared with these of other similar methodologies. A discussion section must be included that will comment on the produced results. Conclusions must summarize the work presented in the paper.
Response 4: New content has been added. According to the new content of the chapter, a new arrangement is made. The methods used in this paper are compared with other methods. The discussion part and the conclusion part have also been revised accordingly. Specific changes can be seen in the new upload document.
Round 2
Reviewer 1 Report
Some comments have been addressed. I have some follow-up comments:
Response 1: Necessary changes have been made to the corresponding content.
Follow-up comment: Introduction is considered separately to abstract. Therefore, DG requires to be defined again in Introduction, paragraph 1.
Response 4: The missing section is mainly due to typographical errors. This revision adds a new content, and according to the new text, new arrangements have been made for the whole article section. You can refer to the new uploaded document for the corresponding content of the modification.
Follow-up comment: The following section and subsection are linked together “2. Characteristics of Typical Intermittent Distributed Generation2.1 Uncertainty Model of Wind Power 102 Generation and Photovoltaic Power Generation”.
Response 5: Detailed definitions are given for each parameter in the formula. Specific parameter settings, data sources and equipment models are added to the validity test and case analysis respectively. You can refer to the new uploaded document for the corresponding content of the modification.
Follow-up comment: It is suggested to discuss the details of the figures immediately after mentioning “Figure x is …”.
That is, “Figure x is …”, follows by the description of Figure x. Figure x+1 is, follows by the description of Figure x+1.
Response 8: The clustering number has been studied in relevant literature. https://kns.cnki.net/KCMS/detail/detail.aspx?dbcode=CJFQ&dbname=CJFD2014&filename=XTLL201409027&v=MDE5MzZZNFI4ZVgxTHV4WVM3RGgxVDNxVHJXTTFGckNVUkxPZmJ1UnRGeTdrVkw3S1BUbkhZckc0SDlYTXBvOUg=
You can see it in the manuscript in the link above. If you feel it necessary, I will do some research and discussion in my manuscript in the next revision.
Follow-up comment: Please omment on the feasibility of calculating optimal number of clusters by CH index, DB index and SH index.
Response 9: Corresponding content has been revised in the new manuscript.
Follow-up comment: The quoting of reference [25] is missing in main paragraph.
Author Response
Response to Reviewer 1 Comments
Point 1: Response 1: Necessary changes have been made to the corresponding content.
Follow-up comment: Introduction is considered separately to abstract. Therefore, DG requires to be defined again in Introduction, paragraph 1.
Response 1: It has been modified.
Point 2: Response 4: The missing section is mainly due to typographical errors. This revision adds a new content, and according to the new text, new arrangements have been made for the whole article section. You can refer to the new uploaded document for the corresponding content of the modification.
Follow-up comment: The following section and subsection are linked together “2. Characteristics of Typical Intermittent Distributed Generation2.1 Uncertainty Model of Wind Power 102 Generation and Photovoltaic Power Generation”.
Response 2: The corresponding chapters have been adjusted.
Point 3: Response 5: Detailed definitions are given for each parameter in the formula. Specific parameter settings, data sources and equipment models are added to the validity test and case analysis respectively. You can refer to the new uploaded document for the corresponding content of the modification.
Follow-up comment: It is suggested to discuss the details of the figures immediately after mentioning “Figure x is …”.
That is, “Figure x is …”, follows by the description of Figure x. Figure x+1 is, follows by the description of Figure x+1.
Response 3: I The corresponding contents have been revised in accordance with the recommendations. You can see it in the new manuscript.
Point 4: Response 8: The clustering number has been studied in relevant literature. https://kns.cnki.net/KCMS/detail/detail.aspx?dbcode=CJFQ&dbname=CJFD2014&filename=XTLL201409027&v=MDE5MzZZNFI4ZVgxTHV4WVM3RGgxVDNxVHJXTTFGckNVUkxPZmJ1UnRGeTdrVkw3S1BUbkhZckc0SDlYTXBvOUg=
You can see it in the manuscript in the link above. If you feel it necessary, I will do some research and discussion in my manuscript in the next revision.
Follow-up comment: Please omment on the feasibility of calculating optimal number of clusters by CH index, DB index and SH index.
Response 4: The commonly used evaluation index of optimal clustering number has been discussed. You can see it in the new manuscript.
Point 5: Response 9: Corresponding content has been revised in the new manuscript.
Follow-up comment: The quoting of reference [25] is missing in main paragraph.
Response 5: It has been modified.

Reviewer 2 Report
The authors have conducted the requested changes.
The paper has significantly improved.
It can be accepted for publication.
Author Response
Point 1: The authors have conducted the requested changes.
The paper has significantly improved.
It can be accepted for publication.
Response 1: Thank you very much for your recommendation.
